# The Effect of Energy Metabolism up to the Peak of Lactation on the Main Fractions of Fatty Acids in the Milk of Selected Dairy Cow Breeds

**DOI:** 10.3390/ani11010112

**Published:** 2021-01-07

**Authors:** Krzysztof Młynek, Agata Danielewicz, Ilona Strączek

**Affiliations:** Faculty of Agrobioengineering and Animal Sciences, University of Natural Sciences and Humanities in Siedlce, 08-110 Siedlce, Poland; agata.danielewicz@uph.edu.pl (A.D.); ilonastraczek1984@gmail.com (I.S.)

**Keywords:** cattle, peak of lactation, lipolysis, fatty acids, casein

## Abstract

**Simple Summary:**

The metabolism of cows up to the peak of lactation significantly influences production parameters and the quality of the production cycle. The aim of this study was to analyze the energy metabolism of selected breed groups of cows and its variability in different stages of early lactation. The analysis was performed using data on the following parameters: body condition score (BCS), fatty acid (FA) fractions, basic milk constituents, and serum parameters (BHBA, glucose, and leptin). Holstein-Friesian (HF) cows and HF crossbreds with Black-and-White Lowland (BW) bulls generally had higher daily yields and reached the peak of lactation earlier. These cows, however, suffered the greatest loss in BCS, which led to higher levels of non-esterified fatty acid (NEFA) in the blood than in the other groups. Cows entering the peak of lactation with less intensive production were shown to have lower levels of leptin and higher glucose concentrations in the blood. The smaller loss of BCS in these cows did not lead to rapid lipolysis, and therefore the release of large amounts of non-esterified fatty acids and β-hydroxybutyrate into the blood was limited. The milk of breed groups that had somewhat lower yield and reached the peak of lactation about 11 days later (Simmental and Black-and-White Lowland) had a more beneficial fatty acid profile and casein content. The results indicate that one of the important factors influencing the intensity of lipolysis is leptin, which regulates appetite. The Simmental and Black-and-White Lowland groups had lower concentrations of leptin. This suggests the need for further research on how appetite and the prolongation of feed intake is linked to longer availability of nutrients, including glucose.

**Abstract:**

During early lactation in dairy cows, metabolic processes are adopted to provide energy and nutrients for the synthesis of milk compounds. High milk production potential includes sudden changes in energy metabolism (negative energy balance (NEB)) that can induce uncontrolled lipomobilization and high blood free fatty acid (FFA) levels. Destabilization of cows’ energy may interfere with endocrine homeostasis, such as the secretion of leptin, a co-regulator of the appetite center. Therefore, it is important to analyze the physiological aspects of the maintenance of energy homeostasis in various dairy breeds. Usually it is crucial for the health of cows, influences the production cycle and lifetime yield, and determines the profitability of production and milk quality. The aim of this study was to analyze the energy metabolism of selected breed groups of cows and its variability in different stages of early lactation. The analysis was performed using data on the following parameters: body condition score (BCS), fatty acid (FA) fractions, basic milk constituents, and serum parameters (BHBA, glucose, and leptin). These results were analyzed in relation to parameters of energy metabolism during the stage up to the peak of lactation. An earlier peak of lactation was shown to be conducive to an increase in the content of non-esterified fatty acids (NEFAs) and of casein and κ-casein. During the study period, parameters characterizing the maintenance of energy homeostasis were usually lower in the Simmental and Black-and-White Lowland cows. Compared to the group with the highest production, their yield was from 2.8 to 4.7 kg lower, but the milk had a more beneficial fatty acid profile and nutrient content, determining suitability for cheese making. At the same time, they had lower levels of NEFAs and β-hydroxybutyrate in the blood, which indicates less spontaneous lipolysis of fat reserves. Concentrations of the appetite regulator leptin in the blood were correlated negatively (*p* ≤ 0.05) with the glucose concentration (–0.259) and positively with NEFA (0.416). The level of NEFAs was at the same time positively correlated with the content of saturated fatty acids in the milk (0.282–0.652; *p* ≤ 0.05). These results contribute to our knowledge of the effect of production intensity on the maintenance of homeostasis up to the peak of lactation in dairy breeds with differing production potential. In practice, this may increase the possibilities of improving milk quality and the profitability of production.

## 1. Introduction

Research on improvement of the fatty acid (FA) profile of milk indicates that apart from the yield of cows, the FA profile is largely determined by the availability of substrates in the diet, fermentation processes in the rumen [1], changes in FAs in the mammary gland [2], and the seasonality of production [3,4]. A significant proportion of substrate transformations, about 85%, involve ester hydrolysis and biohydrogenation in the rumen [5]. The main product of these transformations is energy derived from acetate and β-hydroxybutyrate (BHBA) [1,6]. Up to the peak of lactation, the level of energy is usually insufficient. This induces spontaneous and excessive lipolysis of fat tissue [7,8] and is conducive to the production of non-esterified fatty acids (NEFAs) [9]. Synergy of endogenous factors (negative energy balance, glucose level, and leptin secretion) usually antagonizes feed intake and intensifies the excessive production of NEFAs and BHBA in the liver [10,11,12]. This mechanism involves the appearance of a negative energy balance (NEB) and is linked to the physiological demands of maintaining body functions and increased milk production. According to Drackley et al. [13], these factors can increase the demand for nutrients threefold. The energy deficit increases the mobilization of reserves of energy deposited in the adipocytes. This induces lipolysis of adipose tissue [8] and causes a sharp decline in BCS, which is conducive to the release of oleic acid from the adipocytes [9,11]. Research by Ducháček et al. [14] and Vernon [15] shows that this process need not adversely affect the nutritional value of milk, but it does affect the overall health of cows [14]. Excessive lipolysis of adipose tissue is conducive to the production of non-esterified fatty acids (NEFAs) [9], probably because the FA profile of milk is determined at many levels. Apart from bacterial metabolism in the rumen, FAs are also synthesized de novo in the udder. Lipogenesis here is mediated by stearoyl–CoA desaturase, which mainly affects FAs containing 14 to 18 carbon atoms [16]. Entering the mammary gland together with the blood, these acids affect the quality of milk fat [2]. After the peak of lactation, the energy generated promotes the synthesis of short- and medium-chain fatty acids [2,9]. These acids can undergo further modification in the udder. This process mainly requires β-hydroxybutyrate and acetate derived from fermentation in the rumen [17]. Previous research has focused primarily on environmental factors and the possibility of modifying the FA profile through diet and feed quality. There is little information, however, linking the energy metabolism and productive potential of dairy breeds with the nutritional properties of the milk they produce. In production practice, up to the peak of lactation, this information can be valuable, providing the possibility to improve dairy herds and increase the suitability of milk for processing. The aim of this study was thus to analyze the energy metabolism of selected breed groups of cows and its variability in different stages of early lactation. The analysis was performed using data on the following parameters: BCS, FA fractions, basic milk constituents, and serum parameters (BHBA, glucose, and leptin).

## 2. Materials and Methods

The study was carried out on four breed groups of dairy cows: Holstein-Friesian (HF) cows with more than 75% genes of the breed (20 cows); Simmental (SIM) (20 cows); Black-and-White Lowland (BW), a conserved breed (20 cows); and HF×BW crossbred cows (50:50%—20 cows). The average milk production was characterized by the mean (least-squares method—LSM) and standard error (SE) and age of the cows (LSM ± SE) in each group were as follows: for HF, 8450 ± 42 L and age 2.7 ± 0.21; for SIM, 7342 ± 42 L and age 2.8 ± 0.31; for BW, 6253 ± 35 L and age 3.1 ± 0.52; and for HF×BW, 7235 ± 38 L and age 2.6 ± 0.81. The cows were from four farms and constituted separate experimental groups. The animals were evaluated on average from the 6th day postpartum. The experiment was completed after 110 days of lactation on average.

### 2.1. Feeding and Housing of Cows

The HF and HF×BW cows were kept in pens with separated resting boxes, whereas the cows in the other groups (SIM and BW) were tethered. The cows had direct access to feed and water (open drinkers), and their living conditions met the requirements of good production practice.

Nutrient requirements were established based on information about feed quality [18], the approximate body weight of the breed groups, and forecast milk production for the analyzed stages of lactation (SLs): days 5 to 30 (SL I), days 31 to 60 (SL II) and days 61 to 99 (SL III). The nutritional value of the feed was determined by chemical analyses performed a few days prior to each stage of lactation. Percentages of particle sizes in the feed were determined at the same time. The body condition score (BCS) of the cows was also assessed before each stage of lactation, using a five-point scale. The average from three independent scores was calculated. The BCS was used as an indicator for NEB.

Nutrient requirements and balancing of the diet were determined according to feeding standards for ruminants [19] and INRAtion software. HF and HF×BW cows were fed in a total mixed ration (TMR) system, with the ingredients mixed in a feed wagon. The SIM and BW groups were fed in a partially mixed ration (PMR) system, in which the feed components were placed directly on the feed platform and mixed. In all groups the cows were fed three times a day (on average every 8 h), using feed pushing. The composition of the diet was similar in all herds (Table 1).

The average amount of uneaten feed in each group was determined based on weighing five times a month. On this basis, the average dry matter intake (DMI) was calculated. The average percentages of particle sizes in the diet (PSPS sieves) were as follows: >19 mm (7%), 8–19 mm (52%), 4–8 mm (19%), and ≤4mm (22%). About two weeks before calving, the cows received a preparatory diet. The animals did not show any disease symptoms, e.g., perinatal diseases, ketosis, or acidosis, indicated by milk urea concentration or blood BHBA. To rule out the effect of differences in the diet and housing conditions (farms), a cluster analysis was performed, taking into account the actual energy intake and DMI, considered to be the main factors inducing a negative energy balance (NEB). The analysis showed a high similarity between the farms for these parameters: 0.682 to 0.921 in the case of housing and 0.589 to 0.729 for the diets used on the farms.

### 2.2. Sample Collection and Analyses

Milk samples of about 250 mL were collected using a calibrated milk meter (DeLaval) that simultaneously measured the amount of milk. Milk was collected on average on the 7th day after the start of each SL and 7 days before the end of each SL. In total, 480 milk samples were collected (2 samplings × 3 SLs × 20 cows × 4 breeds). Milk from morning and evening milking was mixed into one sample. The samples were stored in refrigerated conditions (4 °C ± 0.5 °C). The content of protein, fat, lactose, dry matter, and urea in the milk was determined using the Bentley Combi 150 (Bentley Instruments, Chaska, MN, USA). The analysis was performed in a laboratory accredited by the Polish Accreditation Centre.

The content of caseins was determined in skimmed milk (after centrifuging). Total casein was determined using the Kjeldahl method [20] after separating and purifying the precipitate (casein) in buffer with pH 4.6. The content of κ-casein was determined by HPLC (Varian Inc., Palo Alto, CA, USA). Preparation of the matrix additionally involved dissolving the purified casein precipitate in buffer with pH 8.0 and filtering it (nylon syringe filter; 0.25 µm/0.45 µm; Alchem, Poland). Casein proteins were separated on an XB-C18 column (250 × 4.6 mm; Aeris Widepore, Phenomenex, US: 708385-2 S.N.) using carrier phases A—TFA/Acetonitryl (0.1N) and B—TFA/H_2_O (0.1N).

The FA (fatty acid) profile of the milk fat was determined using GLC;Agilent 6890N (Agilent Technologies., Wilmington, DE). The following FA groups were distinguished: short-chain fatty acids (SCFAs), long-chain fatty acids (LCFAs), saturated fatty acids (SFAs), monounsaturated fatty acids (MUFAs), polyunsaturated fatty acids (PUFAs), and unsaturated fatty acids (UFAs). The share of non-esterified FAs (NEFAs) was based on the sum of C16:0, C18:0, and C18:1 cis-9. Fat was extracted using the Röse-Gottlieb method [21]. Transmethylation of FAs to methyl esters (FAME) was at 70 °C ± 0.5 °C (Thermo heat block). GLC modules: autosampler, split/splitless injector (split 1:5), flame ionization detector (FID). Separation was carried out on a 100-m, 0.250-mm column (HP-88; SN:UST458414H, Agilent Technologies Inc., US). Temperature program: injector and FID −250 °C; furnace −95 °C (5 min), 120 °C (15 °C/min for 15 min), 210 °C (25 °C/min for 30 min), 250 (20° C/min for 5 min). Carrier gas flow (He): 5.0 mL/min. The identification of FAs and determination of their percentages were based on retention times (reference Supelco 37.No:47885-U; Sigma Aldrich) in Agilent Tech GC Chemstation A09.03 software (Agilent Technologies., Wilmington, DE). The content of oleic (C18:1) and palmitic (C16:0) acid were the basic markers of a negative energy balance (NEB).

Blood for analysis was drawn before morning feeding from the jugular vein. Due to the potential effect of stress on milk yield, blood was taken 24 h after milk was collected for analysis. Blood was collected twice in each stage of lactation, for a total of 480 blood samples (2 × 3 × 80). Test tubes with sodium fluoride and sodium heparin (Medlab-Products Ltd.) were refrigerated. Blood for glucose determination was placed in ice. Glucose content in the blood was measured using original Randox kits (Randox Laboratories Ltd., Crumlin, UK) and a UV-Vis spectrophotometer (Varian Inc., Palo Alto, CA, USA). Samples for determination of BHBA (β-hydroxybutyrate) were centrifuged at 1500× *g* at 4 °C for 20 min. The supernatant was collected and stored at −75 °C until BHBA analysis using original Randox kits (Randox Laboratories Ltd.) and a UV-Vis spectrophotometer (Varian Inc., Palo Alto, CA, USA). The plasma samples were analyzed for leptin levels using a bovine-specific ELISA kit (EIAab, Wuhan, China). All samples were measured in triplicate.

### 2.3. Statistical Analysis

Statistical analysis of the results was performed in Statistica 13.0 software. Analysis of variance was carried out in a general linear model (GLM) with repeated measures. The effect of diet and housing was verified by cluster analysis, using the k-means method. The model included account the actual energy intake and DMI of the groups. The results are presented as means (least-squares method - LSM) and standard error of the mean(SEM). The significance of differences between means was estimated by Duncan’s test at *p* ≤ 0.05. Correlations (r) between selected parameters were estimated using the Pearson correlation model (*p* ≤ 0.05).

## 3. Results

HF and HF×BW cows reached the peak of lactation on average 11 days earlier than SIM and BW cows. At the same time, they achieved higher daily milk production (DMP), on average by 2.62 kg in the case of HF×BW and 4.65 kg for HF (*p* ≤ 0.05; Table 2).

The table presents body condition scores (BCS) and BCS loss (%) as an indicator of the degree of mobilization of lipolysis. HF and HF×BW, with greater production potential, were found to have greater losses of body condition. In these groups the BCS loss was 21.5% and 24.2%. The differences in relation to groups SIM and BW ranged from 4.75% to 7.45% (*p* ≤ 0.05). The milk of HF and HF×BW cows contained on average 10.5 mmol/L^–1^ more urea. The blood of these groups contains more 0.103 mmol/L^–1^ BHBA and 32.9 µmol/L^−1^ NEFAs (*p* ≤ 0.05). The data (Table 2) show that the HF and HF×BW cows had a higher leptin content in the blood, higher on average by 0.12 ng/mL^−1^ (*p* ≤ 0.05). Compared to SIM and BW cows, HF and HF BW blood contained less glucose, lower on average by 0.18 mmol/L (*p* ≤ 0.05).

The milk of the breed groups did not differ substantially in terms of total protein content (TP), though it was slightly lower in HF (Table 3). Differences were noted, however, in the content of nutrients determining suitability for cheese making. The milk from the HF and HF×BW groups had less casein than that of SIM and BW, on average by 1.4 p.p. (*p* ≤ 0.05), and from 0.014 p.p. to 0.023 p.p. less κ-casein (*p* ≤ 0.05). Their milk usually had a lower content of SCFAs, on average by 0.23 p.p., and higher content of LCFAs, on average by 0.78 p.p. (*p* ≤ 0.05). The content of palmitic acid (C16:0), which usually makes up the largest share of SFA, can to some extent function as one of the markers of NEB at the peak of lactation. The highest level of this acid was noted in the milk of HF cows (26.96%). In comparison with the other genetic groups, in which the content of C16:0 was similar, the differences ranged from 2.49 to 0.83 p.p. (*p* ≤ 0.05). The total content of the SFAs was 0.56 p.p. lower (*p* < 0.05) in the milk of SIM and BW (Table 3).

In the case of MUFAs and PUFAs, the greatest differences were found between the SIM and HF groups (*p* ≤ 0.05). The analysis of MUFA took into account oleic acid (C18:1), which is the main deposit of adipocytes of fat reserves. It is considered a good marker of NEB. Its content was highest in the milk of HF cows (2.86%) and lowest in the milk of BW and SIM, 0.5 p.p. lower (*p* ≤ 0.05) on average. This may indicate that a greater reduction in BCS was accompanied by more intensive lipolysis. The percentages of UFA fractions were lower in the case of the milk from the SIM groups. Compared to the milk of HF cattle, their content was 0.97 p.p. higher on average (*p* ≤ 0.05). The milk of HF cows had the highest SFA/UFA ratio (Table 3). The difference was greatest in comparison to the milk of SIM cows, at 0.11 (*p* ≤ 0.05). The milk of SIM cows also had one of the lowest values for cholesterol content.

A significant factor influencing the metabolism of cows throughout lactation is the dynamics of milk secretion up to the peak of lactation, and the animals’ ability to maintain energy homeostasis is an important aspect of lactation persistence. Up to the peak of lactation, an upward trend was observed in the content of urea in the milk (Table 4). The average difference between stages of lactation (SL) was 14.5 mmol/L^–1^ (*p* ≤ 0.05). In the initial stage of lactation (up to SL II), the more intensive increase in daily milk production (DMP), amounting to 8.5 kg/day (*p* ≤ 0.05), was accompanied by lower production of BHBA (0.362 mmol/L^–1^; *p* ≤ 0.05). However, somewhat less intensive lipolysis of fat reserves was noted during this stage, as indicated by the difference in the content of NEFAs in the blood, amounting to 50.4 µmol/L^−1^ (*p* ≤ 0.05). The more intense lipolysis during this stage is confirmed by the body condition results. The differences noted between stages of lactation were greatest up to SL II. They averaged 0.39 BCS points and were 0.18 points greater than in the later stage (*p* ≤ 0.05).

In the period approaching the peak of lactation, in which the rate of increase in DMP was lower (3.7 kg/day; *p* ≤ 0.05), the level of BHBA in the blood was higher, on average by 0.528 mmol/L^–1^ (*p* ≤ 0.05). Interestingly, the level of NEFAs in the blood during this stage decreased on average by 41.3 µmol/L^−1^ (*p* ≤ 0.05). This was most likely due in part to the effect of leptin, an appetite regulator, of which the level in the blood increased more rapidly up to SL II (0.22 ng/mL^−1^; *p* ≤ 0.05). During this time the synergistic effect with the decreasing level of glucose in the blood, on average 0.18 mmol/L (*p* ≤ 0.05), may have been conducive to lipolysis of fat reserves and production of NEFAs. After SL II, however, when the increase in NEFAs was less dynamic, the level of glucose as a co-regulator of energy homeostasis was relatively stable (0.05 mmol/L; *p* ≤ 0.05). 

Analysis of the values of the NEB markers used in the study revealed the greatest loss of BCS points (Table 4) up to SL II (0.39 points, *p* ≤ 0.05). This was accompanied by a larger increase in the share of C16:0, on average by 2.02 p.p., and C18:1, on average by 1.27 p.p. In the next 3rd SL, the loss of BCS was smaller, on average by 0.21 points (*p* ≤ 0.05), and the increase in the content of C16:0 and C18:1 was 0.73 and 0.83 p.p., respectively (*p* ≤ 0.05; Table 5).

Up to the peak of lactation no significant changes were noted in the protein content in the milk, which was 3.42% on average (Table 5). From SL II, however, the percentage of casein was on average 1.4 p.p. lower (*p* ≤ 0.05). A similar tendency was observed for fat, but the difference was much smaller (0.06 p.p.; *p* ≤ 0.05). It should be noted that in the stage between SL II and III, there was also a reduction in leptin and glucose in the blood (Table 4). In the initial stage after calving (up to SL II; Table 5) there was mainly an increase in the level of LCFAs, by 2.74 p.p. on average (*p* ≤ 0.05), during this stage, whereas level of SFAs increased on average by 2.51 p.p. (*p* ≤ 0.05). The changes were accompanied by an increase in the amount of NEFAs and BHBA released into the blood (Table 4), which explains the higher content of SFAs in the milk obtained after SL II. Up to the second stage of lactation (SL II), the greatest change was noted in MUFAs, of which the content decreased on average by 2.41 p.p. (*p* ≤ 0.05). In the case of PUFAs, although the trend was similar, the average differences amounted to just 0.1 p.p. (*p* ≤ 0.05; Table 5). Differences between SLs were also shown for n-3 and n-6 PUFA content (*p* ≤ 0.05; Table 5). However, no significant differences were shown for the n-6/n-3 ratio. The share of these acids decreased, although only slightly, with the approach of the peak of lactation. The results indicate that the SLs significantly influenced the reduction in UFA content. Their share was lowest in SLs II and III. Compared to SL I, it was on average 2.51 p.p. lower (*p* ≤ 0.05; Table 5). The SLs were not shown to influence cholesterol content. The data in Table 5 also did not indicate any influence of SL on the ratio of SFA to UFA and MUFA.

The correlation coefficients (r) presented in Table 6 indicate that daily milk production (DMP) and day of lactation (DL) are positively correlated with indicators of the intensity of spontaneous lipolysis. Their values with NEFAs were 0.906 and 0.788, respectively (*p* ≤ 0.05), and the correlations with BHBA were 0.777 and 0.885 (*p* ≤ 0.05). The NEFA × leptin correlation (0.416; *p* ≤ 0.05) indicates that a higher level of the appetite-reducing leptin accompanied a higher concentration of NEFAs released during the energy deficit. Moreover, there was also a negative correlation with glucose (−0.259; *p* ≤ 0.05) and glucose with NEFA (−0.386; *p* ≤ 0.05). 

The correlation coefficients obtained for SFAs with parameters of lipid metabolism indicate a negative effect of DMP and DL. The increasing milk production up to the peak of lactation was positively correlated with the content of SFAs, which was confirmed by r values from 0.289 to 0.746 (*p* ≤ 0.05; Table 6). The content of SFAs was correlated positively with the NEFA concentration in the blood (0.274 to 0.689; *p* ≤ 0.05) and was usually negatively correlated with that of glucose (−0.287 to −0.380; *p* ≤ 0.05). In the case of UFAs, negative correlations were noted with DMP and DL, from −0.258 to –0.761 (*p* ≤ 0.05). While the significant relationship between BCS and DMP and BHBA and NEFAs are confirmed by the negative correlation ranging from −0.640 to −0.686 (p ≤ 0.05).

UFAs were negatively correlated with NEFAs, from −0.145 to −0.687 (*p* ≤ 0.05), and with BHBA, from −0.231 to −0.708 (*p* ≤ 0.05). Negative correlations were also shown between UFAs in the milk and leptin in the blood: from −0.158 to −0.352 (*p* ≤ 0.05). Palmitic acid (C16:0) and oleic acid (C18:1) were found to be correlated negatively with BCS and positively with BHBA (Table 6). However higher values for these correlations were found for C18:1.

## 4. Discussion

Selection for milk yield increases the demand for energy and intensifies energy metabolism in dairy cows. One important factor in this case is the changes in the negative energy balance (NEB) and content of nutrients in the milk. In this regard, concentrations of NEFAs esterified to triglycerides and β–hydroxybutyrate (BHBA), as well as the release of C18:1 from the adipocytes, remain the basic parameters of the metabolic profile of dairy cows. Their content is linked to maintenance of energy homeostasis, which is associated with the synthesis of FAs in the rumen and mammary gland [22]. In the early lactation stage, increasing production is conducive to the appearance of NEB. Adipocytes then release FAs that are usually unfavorable for the nutritional quality of milk fat. Ducháček et al. [14] showed that NEB increased the share of hypercholesterolemic FAs in the milk of HF cows by 1.86 p.p. on average. Their study showed that NEB mainly leads to an increase in MUFA, in which case the difference was 1.81 p.p. In the case of UFA the difference was smaller, at 0.33 p.p. A similar tendency was observed in our study, but the milk of HF cows had higher content of MUFA and lower content of PUFA than in the study cited. The milk of the HF breed also had the highest content of C16:0 and C18:1, which is the main deposit in the adipocytes. This may also suggest that diet did not directly influence the FA profile in the milk. Additionally, Sobótka et al. [23] showed a significant influence of lactation phase and breed on the content of FAs. However, significant correlations were found only in the case of SFAs and PUFAs. This did not fully coincide with our results. Sobótka et al. [23] the lowest content of C16:0 was found in the milk of HF cattle. More C16:0 has been found in the Jersey breed and its HF hybrids. However, there was no effect associated with lactation stage. However, it is interesting that the C16:1 released from adipocytes appeared in greater amounts in the later stages of lactation. However, this can be explained by the greater availability of energy for C16:0 synthesis in the mammary gland de novo [16]. Incomplete compliance of our results with the studies by Sobótka et al. [23] may be explained by the research of Poulsen et al. [24]. These researchers explained that environmental variation affects individual FAs of milk differently, based on both breed and feed quality. The dietary effect on FA composition is obvious, but as they suggest, an important factor is also the source of origin of the feed ingredients. Furthermore, Samkova et al. [5] suggested that the shaping of milk FAs may be affected to a large degree by cow individuality, but also by the stage of lactation. According these studies, breed is a factor of lower significance. Samkowa et al. [5] suggest that biochemical changes, especially biohydrogenation in the rumen, should be studied in greater detail. In light of these studies [5,23,24], the obtained FA profiles of milk in our experiment could have been shaped by other environmental factors, as well as the production potential of the studied breeds and their reaction to the formation of NEB. During early lactation it stimulates the content of non-esterified FAs (NEFAs) generated from lipid tissue increases. It is usually proportional to the size of the energy deficit [25,26]. This leads to an increase in long-chain FAs in the milk, which enter the mammary gland with the blood. Craninx et al. [2] reported a lower content of C16:0 in the milk of Holstein cows in a grazing trial. However, the diet used by the authors, containing hay and silage from grass and maize, did not affect the content of C18:1. This may suggest that the diet does not directly affect the share of C16:0, which is influenced by NEB. Craninx et al. [2] showed that in early lactation, the milk of cows with higher yields and with higher fat content had a lower share of FAs, with less than 15 carbon atoms. In our study, this was confirmed by the positive correlations between the level of long-chain FAs (LCFAs) and C16:0 and milk yield (0.593, 0.632; *p* ≤ 0.05) and NEFA content (0.689 and 0.682; *p* ≤ 0.05). A study by Vanbergue et al. [26] showed that spontaneous lipolysis was mainly dependent on the breed of cows and season of production. The study did not confirm the effect of the feeding intensity or its interaction with breed on the course of lipolysis. Although the authors found that the influence of breed on SFA and MUFA was greater, feeding intensity was also found to have a minor effect. This effect, however, varied depending on the production season. In cows with high postpartum milk production, we noted higher concentrations of BHBA and NEFAs in the blood. At the same time, the blood glucose levels were usually lower.

Lipomobilization in high-producing dairy cows up to the peak of lactation could reach a pathological range, disturbing the liver’s morphological and functional efficiency. Diokovic et al. [12] demonstrated that in the early-lactation cows, there is a rapid increase of fatty acids in the liver. The authors observed growth the lipomobilisation markers, especially the serum β–hydroxybutyrate and free fatty acid concentrations. According to the authors, the liver steatosis was affected, which disturbed the synthesis of hepatocyte. Consequently, it led to weaker concentrations of glucose and an increase in triglycerides. This effect has induced some cellular lesions, as evidenced by significant increases in the serum albumin and bilirubin concentrations. Šamanc et al. [27] showed that an NEB is conducive to excess accumulation of fat in the hepatocytes. This situation may be a major cause of endocrine disorders, including secretion leptin. This may result in impaired gluconeogenesis and reduced glycaemia. In comparison with our results, Vargová et al. [28], who studied the hormone profile of Slovak Pied Cattle, reported higher blood glucose concentrations. However, up to nine weeks postpartum they observed a downward trend in NEFAs and BHBA, accompanied by a decrease in body condition score (BCS). The authors [28] noted a positive correlation between BCS and leptin (0.360; *p* ≤ 0.001). Contrary to the studies by Vargová et al. [28], in our study we showed a negative correlation between BCS and leptin (−0.480; *p* ≤ 0.05) and a positive correlation between daily milk production and leptin (0.417; *p* ≤ 0.05). As demonstrated in our studies, the NEFA × leptin correlation (0.416; *p* ≤ 0.05) indicated that a higher level of leptin led to a higher concentration of NEFAs released during the energy deficit. This may be related to decreased appetite. In consequence, reduced feed intake may result in a lower glucose concentration in the blood. This is explained by the negative leptin × glucose (−0.259; *p* ≤ 0.05) and NEFA × glucose (−0.386; *p* ≤ 0.05) correlations. This may indicate that the level of the appetite regulator leptin is linked to the production potential of the breed. In our study, a lower leptin concentration was usually noted just after calving, but it was higher in the groups reaching the peak of lactation with a higher yield. The obtained values of the correlation coefficient with leptin were relatively low, but they may suggest that the stronger appetite in individuals with lower leptin levels may prolong feed intake. In effect, this may be linked to better energy availability and to energy conversions that are beneficial for FAs. This indicates more rapid induction of lipolysis due to reduced appetite in cows with higher yields [27,28]. Reduced appetite and higher production potential may also explain the lower glucose level in the blood of animals during the peak of lactation and reaching this peak with a higher yield. According to authors such as Lock and Garnsworthy [16], a greater energy deficit associated with milk production may reduce direct absorption of cis–9–C18:1 in the small intestine and limit processes involving desaturase in the mammary gland. Liefers et al. [29] reported a lower leptin content in the blood of cows that produced more milk, as in our study. They also showed lower DMI in cows with a negative energy balance. Our results were not confirmed in this case, as the content of C18:1 as an indicator of NEB was usually lower in cows with lower leptin levels. This relationship is confirmed by the correlation of C18:1 × leptin (–0.428; *p* ≤ 0.05). Buttchereit et al. [30] also showed that strong mobilization of energy from lipid tissue is not only conducive to more intensive production of long-chain FAs, but can also reduce the protein content in milk. This is consistent with the tendency observed by Pupel et al. [31] and with our results regarding levels of κ–casein. Our results for the content of the main FA fractions are in agreement with those reported by Petit and Côrtes [32]. They found the highest BHBA level (on average 748 µmol/L) in milk with a higher content of fat and SFAs. The exception was a group whose diet included ground flaxseed. The milk of this group had the highest content of BHBA (1512 µmol/L) and high levels of MUFAs and PUFAs. In this case, the effect was explained by the greater availability of energy reaching the mammary gland, in which a considerable portion of these fractions is generated de novo. In this group of cows, Petit and Côrtes [32] also showed the highest content of non-esterified FAs in the blood (337 µmol/L), although the production level in this case was one of the lowest (29.5 kg/d). The cows in this group, however, had the lowest glucose level in the blood, which in our study was negatively correlated with NEFA (−0.386, *p* ≤ 0.05). Puppel et al. [31] showed very strong negative correlations between the BHBA level in the blood and the content of CLA–9 and –10. Our results are consistent with those reported by Vanbergue et al. [26], who studied the milk of Holstein and Normande cows up to the peak of lactation. The share of SFAs ranged from 61.8% to 73.5% and that of MUFAs from 23.0% to 34.7%. Vanbergue et al. [26] also noted more intensive lipolysis of fat reserves and 1.1% to 5.6% higher content of cis–9 in the milk of cows fed less intensively. Adamska et al. [4], in comparison to our study, reported a higher content of SCFAs (15.99 g/100g FAs) and a similar content of PUFAs (2.53 g/100g FAs) in the milk of Simmental and HF cows. However, the milk of the Simmental breed usually had a more beneficial composition, and also contained more branched-chain FAs. The milk of HF cows, on the other hand, had the most LCFAs (52.11 g/100g FAs) and MUFAs (26.32 g/100g FAs). The results of our study also correspond with those reported by Król et al. [33], in which the milk of Simmental cows contained on average 30.53% SFA, which was on average 4.1 p.p. less than in the milk of HF cows. In the milk of Simmental cows, Król et al. [33] showed higher levels of MUFAs and PUFAs, at 27.13%and 3.43%, respectively. Such large differences were not observed in our study, which can be explained by the use of a similar diet (TMR) in all breed groups.

## 5. Conclusions

The information gathered here suggests that the metabolic reactions of the breed groups were mainly linked to the level of production at the peak of lactation. Breed groups that reach the peak of lactation earlier and with greater daily milk production (DMP) had a greater loss of BCS, on average by 22.8% (*p* > 0.05), and more pronounced changes in NEB markers. These groups, in contrast with cows with lower DMP and a later lactation peak, had higher values for BHBA, on average by 0.102 mmol/L^−1^, and for NEFA, on average by 21.15 µmol/L^−1^ (*p* ≤ 0.05). These changes were accompanied by a higher content of FAs C18:1 and C:16:0, released during NEB from adipocyte deposits. The relationships linking BCS and DMP with NEB markers were confirmed by the correlation coefficients, which ranged from −0.302 to −0.640 (*p* ≤ 0.05) and from 0.632 to 0.906 (*p* ≤ 0.05), respectively. The significant influence of DMP is also indicated by changes in the NEB indicators in the stage up to the peak of lactation. Despite lower DMP in the first stage of lactation, the values for parameters characterizing NEB, including C16:0, were usually higher. This may indicate that the increases in DMP and NEB are more closely linked to the physiology of the breed groups and mainly result from their production potential. The tendencies observed indicate that factors that should be taken into account are leptin and glucose, co-regulators of appetite and NEB, which were negatively correlated (−0.259; *p* ≤ 0.05). However, changes in these parameters in the analyzed groups and stages of lactation, as well as the correlations with BCS, DMP, and C18:1, indicate that they have a relatively strong influence on NEB and lipolysis. The knowledge obtained on the production potential of the breeds can be used to model production adjusted to the environmental conditions of farms.

## Figures and Tables

**Table 1 animals-11-00112-t001:** Feed components and balancing of the diet of the genetic groups. Sample collection and analytical methods. HF—Holstein-Friesian, BW—Black-and-White Lowland, SIM—Simmental.

Forage/Parameters	Breed Group			
	HF×BW	HF	SIM	BW
Number of cows in the group (n)	(20)	(20)	(20)	(20)
Silage maize (kg)	8.73	3.98	5.04	4.86
Haylage (kg)	3.66	5.06	6.35	5.98
Ground cereals ^1^ (kg)	1.31	0.44	0.87	1.31
Supplementary feed Krowimix 18 DE (kg)	1.34	0.45	0.8	1.07
Ground rapeseeds (kg)	1.41	0.88	1.58	1.58
Straw (kg)	0.42	0.48	0.47	0.54
Hays (kg)	0.81	1.64	1.58	1.64
Mineral compound suplement (kg)	0.15	0.14	0.12	0.15
Limestone (kg)	0.10	0.12	0.12	0.11
Total protein (%)	14.2	13.7	13.2	13.8
Dry matter(%)	48.1	47.9	48.5	48.8
Acid detergent fiber (ADF) (%)	22.1	22.5	20.2	19.9
Neutral detergent fiber (NDF) (%)	34.8	34.4	33.5	32.4
Physically effective NDF (peNDF) (%)	14.1	13.9	13.4	13.8
Uneaten feed(%/day)	16.7	17.1	13.4	15.1
Energy (MJ NEL):				
Request	128.5	132.6	125.8	121.5
Intake	112.6	112.8	115.6	110.9
Balace	–15.9	–19.8	–10.2	–10.6
Dry matter intake (DMI) (kg/day)	15.7	15.1	16.2	15.9

^1^ barley to 50%, triticale to 20%, oats to 20%, rye to 10% of the participation; NEL—net energy for lactation

**Table 2 animals-11-00112-t002:** Average yield at the peak of lactation and parameters of energy metabolism in the breed groups.

Parameters	Breed Group *				
	HF×BW	HF	SIM	BW	SEM
Number of samples (n)	120	120	120	120	
Peak yield (kg)	33.8 ^a^	35.8 ^b^	31.5^c^	30.8 ^c^	1.022
Day of lactation (day)	66 ^b^	62 ^b^	74 ^a^	76 ^a^	2.093
BCS 5th day (point)	2.98 ^a^	2.93 ^a^	3.04 ^b^	3.11 ^b^	0.04
BCS 99th day (point)	2.34 ^b^	2.22 ^b^	2.53 ^a^	2.52 ^a^	0.02
Loss of BCS (%)	21.5 ^a^	24.2 ^b^	16.8 ^c^	16.7 ^c^	0.43
Urea (mmol/L^–1^)	172 ^a^	175 ^a^	164 ^b^	163 ^b^	4.591
BHBA (mmol/L^–1^)	1.072 ^a^	1.069 ^a^	0.987 ^b^	0.950 ^b^	0.005
NEFA (µmol/L^−1^)	242.2 ^a^	268.9 ^b^	237.8 ^c^	231.8 ^c^	5.962
Leptin (ng/mL^−1^)	2.82 ^a^	2.80 ^a^	2.71 ^b^	2.67 ^b^	0.028
Glucose (mmol/L)	2.45 ^b^	2.41 ^b^	2.63 ^a^	2.59 ^a^	0.036

^abc^*p* ≤ 0.05; * number of cows in the groups—20; BHBA—β-hydroxybutyrate; NEFA—non-esterified fatty acids (FAs).

**Table 3 animals-11-00112-t003:** Content of nutrients and main fatty acid fractions in the milk of the breed groups.

Parameters	Breed Group *				
	HF×BW	HF	SIM	BW	SEM
Number of samples (n)	120	120	120	120	
Total protein (TP) (%)	3.41	3.37	3.45	3.42	0.454
Fat (%)	4.26	4.23	4.26	4.22	0.394
Casein (% of TP)	77.6 ^b^	77.2 ^b^	78.9 ^a^	78.7 ^a^	2.418
κ-casein (%)	0.369 ^cb^	0.361 ^c^	0.384 ^a^	0.375 ^b^	0.014
Lactose (%)	5.48	5.49	5.43	5.49	0.626
Dry matter (DM) (%)	13.19	12.97	13.14	13.14	0.939
SCFA (%)	9.52 ^b^	9.61 ^b^	9.82 ^a^	9.77 ^a^	0.051
LCFA (%)	59.71 ^a^	59.91 ^a^	58.78 ^c^	59.28 ^b^	0.127
Palmitic acid C16: 0 (%)	24.47 ^b^	26.96 ^a^	24.13 ^b^	24.43 ^b^	0.542
SFA (%)	69.22 ^a^	69.52 ^a^	68.56 ^c^	69.05 ^b^	0.198
MUFA (%)	28.18 ^b^	27.85 ^c^	28.67 ^a^	28.29 ^b^	1.258
Oleic acid C18: 1 (%)	2.57 ^b^	2.86 ^a^	2.39 ^c^	2.33 ^c^	0.154
PUFA (%)	2.59 ^c^	2.61 ^c^	2.77 ^a^	2.66 ^b^	0.033
PUFA n-3 (%)	0.42 ^a^	0.41 ^a^	0.45 ^b^	0.42	0.008
PUFA n-6 (%)	2.17 ^c^	2.20 ^bc^	2.32 ^a^	2.23 ^b^	0.054
UFA (%)	30.77 ^b^	30.47 ^c^	31.44 ^a^	30.94 ^b^	1.133
SFA/UFA	2.25 ^b^	2.29 ^a^	2.18 ^c^	2.23 ^b^	0.678
MUFA/SFA	0.40	0.39	0.41	0.41	0.043
PUFA n-6/n-3	5.20	5.24	5.17	5.27	0.966
Cholesterol (mg/100mL)	20.53 ^a^	20.32 ^a^	19.61 ^b^	19.89 ^b^	1.319

^abc^*p* ≤ 0.05; * number of cows in the groups—20; LCFA—long-chain fatty acid; SCFA—Short-chain FA; SFA—saturated FA; MUFA—monounsaturated FA; PUFA—polyunsaturated FA; UFA—unsaturated FA.

**Table 4 animals-11-00112-t004:** Milk yield and parameters of energy metabolism in cows up to 100 days of lactation.

Parameters	Stage of Lactation (SL) *			
	I: 5–30	II: 31–60	III: 61–99	SEM
Number of samples (n)	160	160	160	
Daily milk production (DMP) (kg/day)	21.2 ^b^	29.7 ^a^	33.4 ^a^	3.029
Urea (mmol/L^–1^)	152 ^c^	168 ^b^	181 ^a^	5.324
BHBA (mmol/L^–1^)	0.613 ^c^	0.975 ^b^	1.503 ^a^	0.008
NEFA (µmol/L^−1^)	196.9 ^c^	247.3 ^a^	288.6 ^b^	3.634
Leptin (ng/mL^−1^)	2.58 ^c^	2.80 ^b^	2.87 ^a^	0.026
Glucose (mmol/L)	2.61 ^a^	2.43 ^b^	2.39 ^b^	0.029
Body condition score (BCS) (points)	3.02 ^a^	2.63 ^b^	2.42 ^c^	0.124

^abc^*p* ≤ 0.05; * number of cows in the groups—80; BHBA—β-hydroxybutyrate; NEFA—non-esterified FA.

**Table 5 animals-11-00112-t005:** Content of nutrients and main fatty acid fractions in milk up to 100 days of lactation.

Parameters	Stage of Lactation (SL) *			
	I: 5–30	II: 31–60	III: 61–99	SEM
Number of samples (n)	160	160	160	
Total protein (%)	3.42	3.39	3.42	0.538
Fat (%)	4.28 ^a^	4.23 ^b^	4.22 ^b^	0.428
Casein (% of TP)	79.2 ^a^	77.9 ^b^	77.7 ^c^	1.232
κ-casein (%)	0.372	0.370	0.373	0.033
Lactose (%)	5.54	5.53	5.35	0.959
Dry matter (%)	13.26	13.17	12.99	0.879
SCFA (%)	9.84 ^a^	9.80 ^a^	9.41 ^b^	0.082
LCFA (%)	57.59 ^b^	59.87 ^a^	60.79 ^a^	0.191
Palmitic acid C16: 0 (%)	23.41 ^b^	25.43 ^a^	26.16 ^a^	0.432
SFA sum (%)	67.43 ^b^	69.67 ^a^	70.20 ^a^	0.174
MUFA (%)	29.85 ^a^	27.70 ^b^	27.19 ^b^	1.255
Oleic acid C18:1 (%)	1.44 ^c^	2.71 ^b^	3.54 ^a^	0.064
PUFA (%)	2.73 ^a^	2.64 ^b^	2.61 ^b^	0.067
PUFA n-3 (%)	0.44 ^a^	0.42 ^b^	0.42 ^b^	0.011
PUFA n-6 (%)	2.29 ^a^	2.21 ^b^	2.20 ^b^	0.071
UFA (%)	32.58 ^a^	30.34 ^b^	29.80 ^b^	1.121
SFA/UFA	2.07	2.30	2.35	0.653
MUFA/SFA	0.42	0.39	0.40	0.031
PUFA n-6/n-3	5.26	5.21	5.20	0.998
Cholesterol (mg/100mL)	19.58	19.46	20.01	1.289

^abc^*p* ≤ 0.05; * number of cows in the groups—80; LCFA—long-chain fatty acid; SCFA—Short-chain FA; SFA—saturated FA; MUFA—monounsaturated FA; PUFA—polyunsaturated FA; UFA—unsaturated FA.

**Table 6 animals-11-00112-t006:** Correlations (*p* ≤ 0.05) between selected parameters.

Indicators	BCS (points)	Urea (mmol/L)	BHBA (mmol/L)	Fat (%)	DL (day)	DMP (kg/day)	NEFA (%)	Leptin (ng/mL^−1^)	Glucose (mmol/L)
BCS (points)	-	–0.364	–0.687	–	–0.715	–0.640	–0.653	–0.480	0.227
DL (day)	–0.715	0.519	0.885	–	–	0.785	0.788	–0.441	–0.341
DMP (kg/day)	–0.640	0.552	0.777	–0.233	0.799	–	0.906	0.417	–0.399
NEFA (%)	–0.654	–	–	–	–	–	–	0.416	–0.386
Leptin	0.480	–	–	–	–	–	–	–	–0.259
SCFA (%)	0.216	–	0.282	0.323	0.289	0.399	0.274	–	–
LCFA (%)	–0.567	0.442	0.595	0.478	0.604	0.593	0.689	–	–0.380
Palmitic acid (%)	–0.302	0.308	0.381	–0.234	0.367	0.632	0.682	–0.293	–0.289
SFA (%)	–0.567	0.486	0.652	0.398	0.714	0.746	0.582	–	–0.287
MUFA (%)	0.558	–0.487	–0.690	–0.299	–0.733	–0.758	–0.682	–	0.292
Oleic acid (%)	–0.605	0.541	0.877	–0.232	0.789	0.871	0.865	–0.428	–0.317
PUFA (%)	–	–	–0.231	–	–0.258	–0.271	–	–	–
PUFA n–3 (%)	–	–	–0.356	–	–	–	–	–	–
PUFA n–6 (%)	–	–	–	–	–	–	–	–	–
UFA (%)	0.568	–0.482	–0.708	–0.324	–0.754	–0.761	–0.687	–0.352	0.293

BHBA—β–hydroxybutyrate; NEFA—non–esterified FA; DMP—daily milk production; DL—day of lactation; LCFA—long–chain fatty acid; SCFA—short–chain fatty acid; SFA—saturated FA; MUFA—monounsaturated FA; PUFA—polyunsaturated FA; UFA—unsaturated FA.

## Data Availability

The data presented in this study are available on request from the corresponding author. The data are not publicly available due to: The research was carried out in private farms. Due to the privacy of farmers, the data will be made available in the form of a database.

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
