# Peer review of "The Effect of Energy Metabolism up to the Peak of Lactation on the Main Fractions of Fatty Acids in the Milk of Selected Dairy Cow Breeds"

_animals, 2021, doi:10.3390/ani11010112_

Round 1

Reviewer 1 Report

The manuscript is worth of being published in Animal, because it improves a current knowledge in metabolism of cows in the early lactation, however there are some lacks in description of material and methods and in disccusion. I think  authors could also focus on selected individual FA such as oleic acid that is the predominant FA in adipocytes and is primarily released through lipolysis during negative energy balance and is thought to be a suitable marker of NEB or palmitic acid that is predominat SFA and contributes largely to changes in SFA during the early lactation. The discussion should be improved and focused on discussing of own findings.

The introduction is too general. The physiological background of FA metabolism and synthesis should be mentioned (e.g. how does high uptake of long-chain FAs by the mammary gland affect de novo synthesis of FAs?) 

L. 64 - 69 - relation of metabolic parameters to BCS should be also mentioned.

L. 85 - 86 - information about milk performance of the herds would be usefull (production per lactation). Were the cows primiparous/multiparous? Parity has an effect on metabolic parameters. Can you provide information about cows BCS before/at parturition and its changes during the experiment? Relationship between BCS and metabolic parameters were proved so this factor should not be omitted, furthermore it can be a way how to express the extent of lipomobilisation. 

L. 87 - 88 - please add information about a length (end) of the experiment.

L. 105 - 115 - description of feeding is not sufficient. Did you analyse the diets? There is no information about sampling and analyses. Please add also dry matter content of the diets. How many times did you evaluate particle size of the diets during the experiment? 

L. 108 - PMR - the abbreviation is not explained, I think this term is not used correctly for described way of feeding 

Table 1 - this term "Ilość niedojadówů should be in English, please unify group designations in text and tables (BW, HFxBW vs CB, HFxCB)

L. 118 - please add milking frequency

L. 120 - 122 - sampling of milk is not clear - milk was sampled from the 8th day postpartum until peak lactation = cca for 2 months? x six times x twice monthly x cca 20 samples per cow ??? - this does not correspond with the no of samples presented in results - 120 per group of 20 cows = 6 samples per animal - please clarify these discrepancies

L. 143 - please add ethical statement for blood sampling from jugular vein in cows (is approval needed?). Why did you decide to sample blood for 6 weeks post partum only when the evaluated period is on average 20 - 34 days longer? Have you sampled all animals in each group? Please clarify this.

L. 166 - 168 and Table 2 - blood and milk results are not "compatible" = blood was sampled only during first 6 weeks post-partum while milk was sampled at least until reaching the peak of lactation, it is 62 - 76 days. Thus the interpretation of blood parameters as representatives of the whole period up to peak of lactation can be misleading.

Table 4 and its interpretation - it is not possible to have data on blood metabolites for all periods of lactation when blood was sampled only during first 6 weeks postpartum - please clarify this!!!

Table 3, Table 5, Table 6 - LCFA instead of LSCFA, 

Table 4 - leptin instead of leptyna, glucose instead of glukoza

Table 5, Table 6 - MUFA instead of PMUFA

L. 284 - 321 - not directly related to own results, relationship between rumen metabolism/variants of TMR etc. and milk and metabolic parameters were not studied

L. 302 - pH

Conclusion is too general.

Author Response

Thank you for your valuable suggestions and comments. Please find attached our responses for your review.

Sincerely

Krzysztof Młynek

Reviewer 2 Report

General remarks:

  • Concerning the conclusions (influencing factors):

The paper presents interesting results – but to my view you should consider feeding (and housing!) as relevant factors, or you should show that they are not present! Why are you sure that feeding/housing is irrelevant? HF had the lowest content of rapeseed in the diet – and a FA-profile which is in accordance to this (additional to high milk yield)! The housing of the “preferred” breeds (tethered) may result in less stress – why didn’t you take this into account? To my view, the presented design does not allow the conclusion “breed is the factor”. Please consider how you can explain the factor “breed” in relation to the bias “housing” and “feeding”!

  • Concerning the guidance of the reader: factors “breed” and “stage of lactation”:

You give Data for “breed” (Table 2 and 3), but later on also for “lactation stage” (Table 4 and 5). I was wondering first, why – and then I thought that “lactation stage” could be used to evaluate breed’s metabolism. I ask the authors to think about, if you want to make two papers out of this data (one with “lactation stage”, and one with “breed/housing/feeding”), or if you can give the reader a clear line to understand WHY you give the data –what is the aim; which argumentative way do you follow?

  • Concerning the guidance of the reader in the discussion:

Your title was quite interesting to me – but in the discussion I did not find what I expected: I supposed to be informed about METABOLISM PROCESSES, and how the findings give additional information asides to basic knowledge. But I found in the most cases only that other authors published comparable results – that was annoying to me. If you want to make the article more interesting, I advise to

1) find a red line to lead the reader!

2) use a clear structure, e.g. a) topic/problem, b) findings/observations, c) explanation

3) be aware of “cause” and “effect” – sometimes your formulation is not precise (example Line 258-259 …” lipolysis is INDUCED by high production”… I would say: attention! High production MAY result in NEB, followed by lipolysis. Your formulation is too much simplified in this context), and in many cases you do not know the real “cause”, you only find simultaneities. You work on “systems”, with several possibilities of reacting on a deficit.

- Concerning the guidance of the reader – congruency between title, aim of the study and results/discussion

The Title sounded interesting – focusing on metabolism and breeds. But the formulation of the aim is not very precise! And the discussion does not represent either the aim nor the expectations due to the title.

I want to ask the authors whether they want to make two papers out of this data – one focusing on lactation, the other on breed, or whether they can rewrite the aim (and results and discussion) to fit to the intention which I think to find behind the presented data:

“to study energy metabolism in early lactation in different herds by milk- and blood-parameters like FAs, cheese-making properties [casein fractions], BHBA, glucose and leptin content, and its variation due to breed (or feed / or housing)”.

If you follow this guide, you could give 1) general information about lactation, and 2) specific variation due to breed / feeding. In the discussion you should look precisely, if you can verify the “breed” as the most important factor, or if you assessed “systems” because you had several factors which were different over the four herds = breeds = housing.

At least you should add a critical review to your design in the discussion and conclusions!

Remarks in detail:

L21: due to the lack of studies … it remains… : WHAT remains relevant?

L27: .. into the blood. The milk… : The linking of the sentences seems unbeneficial for the understanding. The reader may think: now you talk about the other grout, the first sentence was about the “bad” group. Why did you not mention the other breeds (HF+crossbreds)? And be careful with words like “beneficial” – for what, in which case is it “beneficial” – “beneficial” is an evaluation, not a result, and needs to be related!

L34: after reading the simple summary, I had the question: “what is THE REASON for the stable metabolism”?

L56: Why do you use the keyword “production peak”? I would use “peak of lactation”.

L79-81: The aim… : please find a formulation which is more precise! (e.g. as I proposed above).

L80: … energy metabolism PROFILE… : in this study I find “markers”, single substances or sums of them, which you relate to one another. A profile evaluation was done in an outstanding way of statistic evaluation by Baars et al. (2019): Patterns… https://www.mdpi.com/2076-2615/9/3/111 , who used “the FA-profile as a whole” to relate it, in this case to region of origin and season. Think about this possibilities!

L85-86: mention in all cases the unit “cows” (20 cows).

L86: BW is in the most tables named “CB”, and in Table 1, they are noted as n=15! Please check for uniformity!!!

Table 1: - consider if the housing should be mentioned in the table, too!

  • Add a statistic (if possible) to prove that the diets were “similar” (you state that in L110).
  • I see in the diets DIFFERENCES – it would be fine to relate them to the FAs / the metabolism! (especially in the parameters “balance”, but also “cereals” and “rapeseed”, and DMI). On this point I have the question: why did the Hf did not eat as much as the others to retain energy balance? THIS I would like to have discussed later!

L113: Why do give the particle size “on average, and not “per breed/herd”?

L115: how did you evaluate the disease symptoms? à explain in “Methods”!

L121: Why do you report the “average” of days of collection pp?

The sampling sequences are not clear! 6x? each 14 days? (later on: week 1, 2, 3, 4, 5, 6) – and how do you reach 20 samples per cow???? And sampling included day 100 of lactation?

L133: in L 152-153 you tell about the FAs which are summed. Please add the summed FAs also for the other sum-parameters (SCFA, SFA…). And shift the L153 to this part above!

L157: GLM: explain the abbreviation!

(can you add several influencing factors in this software? I would use the factors “herd”, “housing” and “Stage of lactation” (+ a random?) – these I think are important in the design as you presented it and as I understood what you did. Then you may see more appropriate the importance of the influencing factors.

Table 2, 3,  etc: The letters for significance have to be in relation to the scale! Highest values are “a”! lower values are b, c etc.

L177: “more favourable” – in which context! As result you should only say “higher” or “lower”, in the discussion you may evaluate!

Table 3: LSCFA --> LCFA

Table 3 – Table 5: Use the same parameters / the same arrangement in both tables to compare factors “breed” with “lactation”.

Table 5: PMUFA --> PUFA

L213: SCFA = short chain FAs

L244: use uniformity in the explanation: eg. always FAs, not sometimes “fatty acids”.

L250: this is an explanation – I advise to shift it to the discussion “explaining WHY the findings are congruent with knowledge. What do we find NEW in this data?

L255-6: … BHBA remain the basic parameters… : this statement has to be explained. Who said this, why is it like this?

L259: induced by… : at the time /siumltanuously! Or is it really the direct induction?

L254ff – discussion, general remarks:

  • I’m interested in “metabolism / metabolic processes”, not in “results are consistent with others”.
  • The presented date represent “breed” AND “lactation” – it would be fine to find in the discussion the reason for this two aspects! Is lactation a “reference” to evaluate the breeds/herds? I think you could likely do so!

L166: no results in “discussion”.

L274ff: the described content is much too complex! I did not get what you want to explain – the aim of the discussion is missing for me!

L290: the statement “this hypothesis is… explained by…” is just a statement – when you want to lead the reader, you should friendly help him to find the relevant markers, so that he can see by himself in this “picture” which you did draw in his thoughts that the results are congruent e.g. with existing knowledge.

L322: Here I read an accumulation of other results which are “in congruence” with the presented data. Why do you do this? It is difficult for the reader to follow this amount of information – please try to point out the major point you want to explain by this!

L338 – conclusions – general remark:

Please add a critical view on the experimental design, including co-factors biasing the breed!

(references were not evaluated)

Author Response

(The authors gave the same response as above.)

Round 2

Reviewer 1 Report

Authors responded satisfactorily to my comments and suggestions. The manuscript has been significantly improved and can be published in Animals.

I have only one request - please correct throughout the article - oleic acid is abbreviated as C18:1!!! (no C16:1, it is a palmitoleic acid!!!)

Author Response

The C16: 1 designation of oleic acid has been corrected C81: 1. There was a mistake with the automatic conversion in the text. However, it's my fault that I didn't check it and correct it sooner.
Thank you for opinion

Reviewer 2 Report

To the authors:

Your revisions improved the article, the congruency with the title/aim of the study becomes more obvious! And the linking between stage of lactation and breed production intensity is now obvious.

Unfortunately the new paragraphs look like being formulated in a hurry – I’m not pleased by uncareful formulations! Please take you time to return a thoroughly revised manuscript – that is more beneficial for all involved persons.

Remarks in detail:

General:

  1. a) The corrections are missing in many cases the blanks. Please check and correct this in the whole document.
  2. b) The revised manuscript highlights words as „corrections“ which had been already present in the original version – why?
  3. c) In the Tables, the letters to indicate significance are in the presented document not ordered with „high values = a“ and „lower values = b or c“, although you told in your reply to the reviewer that you would have done so. Why is it not changed?

L32: I would formulate the sentence like: DURING the early lactation in dairy cows THE metabolic processes ARE adopted to .... – please check, if this is in your meaning.

L39: Why do you not use a new paragraph for the sentence „.The aim of the study was...“?

L88: Is there a fermentation in the udder??? I suppose you mean „rumen“.

L102: do you mean „the average MILK production“?

L104: for SIM, the SEM for lactation no. is missing.

(you may think about using the term „age“ instead of „lactation number“, or think about using an abbreviation – at least stay consistent in the term)

L115: the abbreviation I would change – please check if you feel that it is useful in your case! My suggestion: „... stages of lactation (SL): days 5 to 30 (SL I), days 31 to 60 (SL II), and days 61 to 99 (SL III).“

L119: I would formulate the sentence like this: „The BCS was used as an indicator for NEB.“ You already explained above that BCS is body condition score, and NEB abbreviation was also given before, you do not have to repeat a once given abbreviation link.         That BCS is „subjective“ is not correct, because you made it at least „intersubjective“ by three independent scorings (evaluated by different persons, I suppose?). „subjective“ is missing scientific basis, I would avoid this and formulate in way that the reader can find that the parameter was evaluated on a validated or reproducible basis. Otherwise it is not worth reporting it.

L121: you tell here „weighing five times a month“, but later on (L134-135) you tell that „Weighing was twice a week...“. Tell it once, and use a formulation which is consistent with the data you use in the statistics.

L133: the word „similar“ sounds inappropriate, I would choose „comparable“ – please ask a native speaker which formulation is appropriate in this case. Later on I found the same word („similar“) and was stopping in reading: it sounds wrong to me.

L138: I would change the sentence as follows: „The animals did not show any disease symptoms like e.g. perinatal diseases or had ketosis or acidosis, indicated by milk urea concentration or blood BHBA." Please check, if this new formulation is correct, otherwise feel free to change or revise otherwise.

L139: first, I supposed that the paragraph has to be shifted to „statistical analysis“, because you explain the statstics more than the evaluted topic. Maybe you want to change the formulation; please check!

L147 – 152: Sampling days are reported in double, and the topics red line is not perfect. I would formulate like: „Milk was collected on average on the 7th day after the start of each SL and 7 days before the end of each SL. In total, 480 milk samples were collected (2 samplings x 3 SL x 20 cows x 4 breeds). Milk from morning and evening milkling was mixed into one sample. The samples were stored in refrigerated conditions (4+- 0.5 °C).

L178: (2 samplings x 3 SL x 20 cows x 4 breeds), or you just say „sampling parallel to milk“, which is already clear because of your formulation before.

L202: I would not say „subjective expression“, but instead „indicator“. Change, if you want to.

L204: „the values“ is a very general word – it would be fine to concretely formulate what you mean: „BCS scores and losses“ is meant, I suppose.

At the end of the line, the numbers are missing a unit , I suppose you mean „BCS loss (%)“.

L207, 208: „on average by 0.18 mmol LESS...   ... leptin: 0.12 ng/ml MORE.“ (less and more may help the reader to understand, or is it wrong what I think?

L210, Table 2:

  • „number of the cows in the group: 20“ – why do you give it in the header? I would replace it below the table, as explanation, marked by *.
  • „BCS 5 day“ sounds as if you report „the first five days“. I think it would be better to use „BCS 5th day“ and „BSC 99th day“ (or „BCS on day 5“ )

L214: ... „althoug it was usually lower in HF“ . This formulation is not very clear. May be you formulate it as „was slighly lower in HF“, then you are correct in respect to non-significant values. Please check and decide if you want to change it.

L222: I would change the sentence to: „The total content of the SFAs was 0.56 pp lower (p<0.05) in the milk of SIM and BW.“ Please check!

L230: you write that UFAs were lower in SIM and BW milk – but this is not in congruence with the data in the table. Also the letters for significante are inconsistent with the values (high values = a). Please check and correct!

Table 3: the order of the letters for significante for MUFA is NOT „a“ for highest values. (I did not check all parameters, I just observed this case  - please check in the other cases).

Table 3: Sometimes you say „Stage of Lactation“, sometimes „Period of Lactation“ – please decide if you want to use both or only one of the terms.

  • „number of cows in the group“ should be replaced below the table, I think. OR you insert this information in the first line „number of samples“ (2 samples per cow), or something like this. (the same for Table 5).
  • Think about the BCS and DMP, if you want to give only the abbreviation or both. Can you add for BCS the unit (points?)

L272: „in the next SL“ – do you mean „in the 3rd SL“ or „in SL III“ ?

Table 5 and Table 3: „total protein“ is in one case abbreviated with „CP“, in the other with „TP“.

Table 6: please add the units for all parameters, or leave it always.

L332: „This breed“ – I suppose you mean HF?

L333: Why do you not make a break in the text flow by using a new paragraph before the new sentence: „During early lactation...“?

L350f / L361 – 364: You repeat the first part; the part below is better formulated. Please revise CAREFULLY the whole part which you inserted! Some sentences are really bad to understand.

L352: „milk growth period“ is a term which I never heard before. Do you mean „increasing milk production“?

L359: you use abbreviations which are not explained. (AST, GGT, LDH).

L370: „our study also reveal... – the ALSO implies that BCS and DMP are evaluated „as the same“ – is it correct to do so?

L393: the correlation coefficient of -0.193 – can you add between which parameters this r-value was obtained?

L415: PHF – what is meant by this abbreviation?

L433-436: This last explanation is to my view the best explanation why the one study ends up in different results than yours. If you can use this view in the discussion part above (especially the new inserted part), then the part becomes more straight and focus on this end, and do not „search“ for an explanation. You may also say (if adequate) that results of Sobota, Poulsen etc. cannot be compared with the presented results becaus a) they had different conditions (feeding intensity related to breed, feed uptake...), b) other reasons. When you can explain why their results are different, than it would be very fine – but I do not expect from you that you can do so, because a lot of knowledge about the surrounding conditions is needed to explain in detail WHY something is not in accordance with „general knowledge“. (I hope you understand what I mean).

L456: the last sentence rises the question in me: How will you do this? It is a new topic, why do you add it here? I would skip it, and maybe you want to add a sentence which focus on milk quality, (FAs, cheese-making), which is a topic you stroke before (e.g. L215).

Separate remark:

The literature „Baars et al.“, which I told you from, is in my view not so relevant for the topic milk quality / season or region influence on FAs, but much more of importance because of it’s statistical way of analysis of the data, because the whole spectra of FAs was used, not single fractions, to evaluate relations... This was meant for you to get an idea of how to analyse data „as profile“ – if you ever would like to do so., what I do not know. The acutal way you are following is the of metabolic pathways, focused on substances linked with one another. The „profile“ evaluation skips this details and leave it in a „black box“, but the relations of factor to effect can be found, too.

Author Response

Thank you for reading our manuscript in detail. This helps a lot in improving the manuscript. I hope you will be satisfied with the changes.
